# Solid-State Dewetting as a Driving Force for Structural Transformation and Magnetization Reversal Mechanism in FePd Thin Films

**DOI:** 10.3390/ma16010092

**Published:** 2022-12-22

**Authors:** Arkadiusz Zarzycki, Marcin Perzanowski, Michal Krupinski, Marta Marszalek

**Affiliations:** Institute of Nuclear Physics Polish Academy of Sciences, PL-31342 Krakow, Poland

**Keywords:** thin films, solid-state dewetting, magnetism, magnetization reversal mechanism, FePd alloy, phase transformation

## Abstract

In this work, the process of solid-state dewetting in FePd thin films and its influence on structural transformation and magnetic properties is presented. The morphology, structure and magnetic properties of the FePd system subjected to annealing at 600 °C for different times were studied. The analysis showed a strong correlation between the dewetting process and various physical phenomena. In particular, the transition between the A1 phase and L1_0_ phase is strongly influenced by and inextricably connected with solid-state dewetting. Major changes were observed when the film lost its continuity, including a fast growth of the L1_0_ phase, changes in the magnetization reversal behavior or the induction of magnetic spring-like behavior.

## 1. Introduction

Fe/Pd multilayers and FePd alloys are of great interest due to their possible applications in magnetic recording [1], permanent magnets [2], surface acoustic wave filters [3], and terahertz magnonics [4]. Such a wide range of applications results from the unique properties of Fe/Pd and FePd systems, which exhibit many interesting magnetic phenomena. In particular, the Fe/Pd bilayer shows the presence of spin-orbit torque [5] and the Dzyaloshinskii–Moriya interaction at the Fe/Pd interface [6] which opens the possibility of skyrmion engineering in this system [7,8]. A small Gilbert damping constant of 0.002–0.004, together with the prospects of its tuning [9], makes the material useful for spin-transfer-torque magnetoresistive random access memory [10,11], or the detection of spin waves [12,13,14]. In the case of ultra-thin Fe_x_Pd_1−x_ alloy, the dependence of magnetic anisotropy on electrical voltage as an indication of the magnetoelectric effect was found [15]. For the alloys enriched with palladium, a spin current polarization appears with Pd atoms acting as highly spin-dependent scattering centers [16]. Furthermore, magnetostriction and magnetic shape memory effects were observed in FePd thin films [17] with potential application as actuators for biomedicine [18,19], photothermal therapy [20] or as a coating layer for titanium implants with good blood- and histo-compatibility [21]. The FePd shows also catalytic [22,23] and sensing properties [24,25] and can be used for the fabrication of ferrofluids with high metallic and thermal conductivity [26]. It has been demonstrated [27,28] that some of these properties, such as hydrogenation capabilities, depend on the magnetism of FePd thin alloy films.

These properties of Fe-Pd systems are strongly related to the crystallographic structure and stoichiometry. For an FePd alloy with an atomic ratio close to 1:1, only the chemically disordered A1 phase and chemically ordered L1_0_ phase are thermodynamically stable at room temperature. The cubic A1 phase is created spontaneously and the transformation to the tetragonal L1_0_ phase with strong uniaxial magnetocrystalline anisotropy of the order of MJ/m^3^ [29] proceeds through an ordering process which progresses gradually in a cascade-type transformation. It involves decomposition into *fcc* phases and congruent transformation to metastable L1_2_, L1′ (modified L1_0_ formed from hybridized L1_0_–L1_2_ phases) or *fct* A6 phases, and then atomic ordering to the L1_0_ phase [30,31,32]. The formation of the L1_0_ phase with an easy axis of magnetic anisotropy normal to the film plane, i.e., perpendicular magnetic anisotropy, requires proper fabrication conditions and thermal treatment performed during or after deposition [33,34]. The polycrystalline film annealing at temperatures higher than 400 °C initiates the formation of L1_0_ crystallites [35] while further annealing at 600 °C leads to a significant increase in L1_0_ phase contribution accompanied by an enhancement of the magnetic coercive field and magnetic anisotropy [36]. The annealing of the thin film is usually accompanied by solid-state dewetting, a process whereby isolated particles are formed while the material remains solid. Its onset is determined by the interface energy between the thin film and substrate [37] and can limit the reliability of devices based on such films due to the loss of film continuity [38]. On the other hand, adequately controlled solid-state dewetting can be used for the fabrication of nanomaterials with new features and properties [39]. In particular, the phenomenon has been used for FePd systems to create free-standing magnetic particles [40], release residual stress [41], fabricate patterned nanoparticles with preferred crystallographic orientation [42], or produce particles with different morphology depending on the used substrate [37]. Additionally, the process can be controlled by doping with other elements, such as Cu, or by substrate nanopatterning to produce an ordered array of nanoparticles [43].

Despite the broad use of solid-state dewetting, its influence on L1_0_ phase formation in FePd systems has not yet been thoroughly investigated. During annealing, polycrystalline FePd alloy thin films exhibit simultaneous phase transformation together with solid-state dewetting. An in-depth understanding of their mutual influence is crucial for the controlled fabrication of FePd thin alloy systems with desired magnetic properties. The relation between these two processes is, therefore, the central point of our studies. We started with the Fe/Pd system and performed annealing at 600 °C at different times, up to 5 h, while tracking the changes in morphology, structure, and magnetic properties of the films. We demonstrated that the phase transition between the A1 and the L1_0_ phase is driven by the solid-state dewetting with strong correlations between morphology, structure and magnetic properties. Those correlations are the main subject of our investigations and show how solid-state dewetting influences chemical transformation and magnetism. In particular, we demonstrated that the most significant changes are observed when the film loses its continuity, resulting in the fast growth of the L1_0_ phase and a change in magnetization reversal behavior, prompting magnetic spring-like behavior.

## 2. Materials and Methods

The samples were prepared with the thermal evaporation method. A multilayer of Fe (4 N, Trace Sciences International) and Pd (5 N, Sigma-Aldrich, Saint Louis, MO, USA) was deposited onto a $Si/SiO_{x}\left(100\;\mathrm{nm}\right)$ substrate in an ultrahigh vacuum (base pressure 10^−9^ mbar). As a result, a multilayered stack of $Si/SiO_{x}/Pd_{1.5\mathrm{nm}}/{\left[Fe_{1\mathrm{nm}}/Pd_{1\mathrm{nm}}\right]}_{5}/Pd_{1\mathrm{nm}}$ with a total thickness of 12.5 nm and Fe_45_Pd_55_ atomic stoichiometry was obtained, and was chosen to fall into the middle of the Fe:Pd ratio range for L1_0_ phase formation [44]. The layer thickness was controlled during the deposition with a quartz monitor and confirmed ex situ with X-ray reflectometry.

The transformation of the Fe/Pd multilayer into a FePd thin alloy film was performed by annealing at 600 °C in a high vacuum of $10^{-5}$ mbar or better. The annealing at high temperatures was performed for different times between 0 and 300 min, where 0 means immediate cooling down of the sample after achieving 600 °C, whereas in other cases, the samples were kept for 4, 15, 30, 60, and 300 min at the final temperature before turning off the heater. The heating ratio was kept at 10 °C /min. As a result, a series of FePd thin alloy films were created.

The X-ray diffraction (XRD) measurements with $\theta_{XRD}/2\theta_{XRD}$ Bragg–Brentano geometry were performed with a PANalytical X’Pert Pro diffractometer using Cu $K_{\alpha }$ line. A detailed description of the measurement procedure for thin film measurements can be found in [45]. The analyses were conducted with FullProf software [46]. Morphology studies were performed using scanning electron microscopy (SEM) (Tescan, Vega 3) with an electron beam energy of 3 keV and with the use of a secondary electron detector.

Studies of macroscopic magnetic properties were focused on field-dependent magnetization curves M(H) and were performed with a Magnetic Properties Measurements System (MPML XL) from Quantum Design company equipped with a Superconducting Quantum Interference Device detector (SQUID). The measurements were performed at room temperature in a magnetic field range of ±50 kOe. The angle of the magnetic field with respect to the sample surface varied between 0 and 90 degrees for all samples. The signal from the holder and the substrate was subtracted from the measured data and only the ferromagnetic part of the signal was analyzed.

## 3. Results and Discussion

### 3.1. Crystallographic Structure and Morphology

The XRD diffractograms for FePd alloy thin films obtained after different times of annealing are presented in Figure 1a. The peaks located around 24, 41, 47 and 49.5 deg show the presence of the FePd L1_0_ phase with tetragonal distortion and correspond to (001), (111), (200) and (002) crystallographic planes, respectively [47]. Both the L1_0_ *fct* phase with space group *P4/mmm* and the *fcc* FePd A1 phase with space group *Fm-3m* are necessary to reproduce the experimental patterns (red and blue colored areas in Figure 1a) [48]. The presence of both phases shows that the samples are a mixture of chemically disordered A1 and ordered L1_0_ phases, that coexist even after 5 h of annealing. The high intensity of (111) Bragg peaks indicates a strong (111) crystallographic texture for both phases.

The lattice parameters for L1_0_ and A1 phases are presented in Figure 1b and collected in Appendix A with the percentage contribution of the L1_0_ phase. The amount of the L1_0_ phase increases with annealing time and the largest rise occurs for the time between 15 and 30 min. The *fcc* disordered phase shows an almost constant and temperature independent lattice parameter *a* of 3.81 Å, close to the bulk value. On the other hand, the *a* and *c* lattice parameters of the L1_0_ phase slowly increase with annealing time (Figure 1b), and the most significant rise again takes place between 15 and 30 min of annealing. This effect is better seen in cell volume dependence on annealing time (Figure 1c), which shows an increase for the L1_0_ phase above the annealing time of 15 min and reaches the bulk value for the samples annealed longer than 30 min. In the case of the sample annealed for 5 h, the values of cell volume of both phases are equal. The tetragonal distortion (the inset of Figure 1c) does not change significantly with annealing time and equals 0.955. Such behavior corresponds to the situation when the *a* and *c* parameters increase similarly in all crystallographic directions, demonstrating the proportional growth of the crystallographic cell. The c/a value is slightly smaller than the value of ~0.965 usually observed for thin films [49] and single crystals [50], indicating a larger tetragonal distortion of the studied films. This could be related to the nonstoichiometric iron to palladium ratio, an effect also observed in the work of Bahamida et al. for Fe_56_Pd_44_ alloy [33].

The XRD data were used for calculating Scherrer coherent length (*L*_coh_), a parameter connected to the crystallographic sheer grain size [51]. Figure 1d presents the *L*_coh_ dependence vs. annealing time calculated for L1_0_ and A1 phases using (111) Bragg maxima. The coherent length shows an increase for initial annealing and tends to saturate for annealing times above 60 min. The data were fitted with the time-dependent grain growth model described as [52]
(1)$$L_{coh}^{n}-L_{0}^{n}=c*\left(t-t_{0}\right)$$
where the parameter $L_{0}$ is the initial grain size; n is the growth exponent; t is the annealing time; $c=c_{0}\exp\left(-E_{a}/k_{B}T\right)$ is a growth rate parameter which has a constant value for annealing at constant temperature T; $c_{0}$ is the initial growth rate constant; $E_{a}$ is the activation energy for grain boundary motion; and $k_{B}T$ is thermal energy. The Scherrer coherence length $L_{coh}$ for the L1_0_ phase shows growth after 30 min of annealing; therefore, an additional initial grain growth time parameter $t_{0}$= 30 min was introduced to reproduce the data for this phase. The obtained values of the n exponent between 9 and 11 for A1 and L1_0_ phases, respectively (see Figure 1d), indicate a very slow grain growth process found previously for FePd [33] or FePt [53] systems.

The L1_0_ phase shows an unusual behavior of crystallographic grain growth. For a short time of annealing, up to 30 min, the crystallographic grains nucleate with a mean size of approximately 10 nm, while only after 30 min the process of grain growth starts. The 30 min of annealing was previously mentioned as the time when a significant increase in the contribution of the L1_0_ phase was noticed (associated with the decrease of the A1 phase contribution). This is a completely different behavior from the A1 phase where the nucleation of crystallographic grains and the growth take place simultaneously with the $t_{0}$ parameter equal to 0.

The values of the coherence length are larger than the thickness of the film, suggesting material dewetting during the annealing process, which was confirmed by the morphology studies using SEM (see Figure 2). For the shortest times of annealing, small voids in the FePd film appear filled with small particles (islands). The size of the voids and the number of particles increase with the annealing time. The most significant change is found in images taken after 30 and 60 min of annealing (Figure 2e,d), where the voids merge and the whole surface is covered only with the isolated particles, and the layer loses its continuity.

The SEM images can be used for quantitative analysis of the FePd lateral particle size distribution. The histograms of the particle sizes are presented in Figure 3a and are fitted with a generalized gamma distribution function [54]. For the short times of annealing, the smallest particles dominate and the histograms show fast decay. On the other hand, for the samples annealed for 60 and 300 min, a maximum in particle size distribution above 200 nm is found, as presented in Figure 3b. The mean particle size shows a rise between 30 and 60 min of annealing, corresponding to the change in film morphology caused by solid-state dewetting. It is connected with a loss of film continuity leading to the reduction of the number of new nucleation centers [55] (see Figure 2d,e). For the longest time of annealing, the FePd alloy is mostly in the form of separated particles (islands) which restrict the ability to nucleate new ones. In that case, a dominant process is the slow growth of existing particles and the reduction in the number of the smallest ones. Similar behavior was found by Barrera et al. for Fe_80_Pd_20_ alloy [40], where the authors observed emerging voids in the film evolving into isolated islands after annealing for 30 min and the successive increase in the islands’ mean size.

The mean particle sizes presented in Figure 3b were fitted using Equation (1). The initial time $t_{0}=30$ min was used, as in the case of crystallographic grains of the L1_0_ phase. The obtained value of the growth exponent of 8.8 shows that in this case, the growth of the islands is also very slow. The growth parameters behave similarly to the coherence length of the L1_0_ phase, suggesting that solid-state dewetting is an important factor necessary to induce the diffusion and transformation of the A1 phase into the L1_0_ structure. The particles (islands) at the beginning of the dewetting process are composed of an A1 and L1_0_ crystallites mixture with a predominance of the disordered phase. When the film losses continuity, the transformation of the A1 phase to the chemically ordered L1_0_ phase proceeds. Structural analyses show that these two processes are inextricably linked.

A closer analysis of SEM images shows that dewetting in the FePd film is a two-step process. At first, the top of the film corrugates and a grainy structure is created while the structure of the bottom part of the film remains unchanged. In the next step, the islands are formed and the substrate is exposed. Similar processes of solid-state dewetting were observed for silver, gold and nickel [56,57,58]. The two-step process is clearly visible in the SEM image of the sample annealed for 4 min at 600 °C (Figure 3c). The bottom part of the image presents a nominal, non-dewetted FePd film having a light grey color. The upper part of the figure shows a region where dewetting caused a partial reduction of the film thickness (a grey color) with small dark areas where a silica substrate is exposed. Finally, the islands formed in the process are visible as white dots. These different dewetting regimes are well illustrated on the profile line where the large particles give high peaks while the partially dewetted film has a value between 0 and 1. This effect can also be found in other samples annealed for various times. A closer inspection of the images (see Appendix A) reveals the presence of areas with different stages of dewetting in all samples. Even the sample annealed for 300 min still shows some regions with a continuous flat film, demonstrating that the process of FePd alloy formation does not reach thermodynamic equilibrium and longer annealing times are needed to improve quality and enhance the contribution of the L1_0_ phase. Indeed, Vlasova et al. showed that even after annealing at 550 °C for 100 h, FePd forms a mixture of different phases [59].

These observed correlations between morphology and crystallographic structure demonstrate that the L1_0_ phase starts to dominate in volume over the chemically disordered or partially ordered phases when the film loses its continuity. Those observations can be compared with the studies described in [31], where two stages of structural transformation were found: a cooperative displacement and transformation from the A1 to the A6 structure, and in the next step, further ordering from the A6 to the L1_0_ phase. Therefore, solid-state dewetting not only leads to material agglomeration, but also supports the transformation of chemically disordered phases to an ordered L1_0_ structure. In this context, the dewetting and diffusion process works simultaneously to create and enhance the formation of the L1_0_ phase.

### 3.2. Macroscopic Magnetic Properties

Field-dependent magnetization curves measured for annealed samples with the magnetic field applied perpendicular (⊥) and parallel (||) to the sample surface are presented in Figure 4. The magnetization saturates at approximately $3\mu_{B}/$Fe for all samples, which is the value expected in FePd alloy [60,61].

For the pristine sample, an in-plane magnetic anisotropy is observed (see Figure 4a), whereas annealing for 300 min results in a similar shape of in-plane and out-of-plane hysteresis loops signifying the isotropic distribution of magnetization local easy axes. The hysteresis loops for intermediate annealing times show a systematic change of magnetic anisotropy from in-plane to almost isotropic behavior.

The parameters extracted from $M\left(H\right)$ curves are presented in Table 1. Coercivity larger than 1 kOe is found in all cases with a tendency to increase with annealing time. It saturates around 2.5 kOe and 2 kOe for in-plane and out-of-plane configurations, respectively (see Figure 5a). The maximal value is reached for an annealing time of 60 min or longer and the obtained values are close to those found in the literature [1,36,62]. The effective anisotropy constants $K_{eff}$, calculated according to the method described in [63], are of the order of $10^{5}$ J/m^3^, similar to the values for thin-film systems or nanoparticles [64]. The magnetic anisotropy values decrease with annealing time.

To verify and precisely determine the direction of the easy and hard axis of magnetization, a series of angle-dependent hysteresis curves were measured for the magnetic field applied at various angles φ with respect to the sample surface. Figure 5b shows the values of magnetization squareness $M_{R}/M_{S}$, where the lines are results of fitting according to the formula:(2)$$M_{R}/M_{S}\left(\varphi \right)\;=A*\cos\left(\varphi -\Delta \varphi \right)+M_{0}$$

The φ parameter is an angle between the applied magnetic field and the sample surface, A is an amplitude and $M_{0}$ is a squareness value for φ−Δφ=90°. The measurements were performed for angles between 0 and 90 deg, so an additional parameter, the phase shift Δφ, is needed to precisely identify the direction of the easy magnetization axis. Results of the fitting procedure with Equation (2) are presented in Table 2. It is seen that the in-plane direction is an easy axis of magnetization, regardless of the applied annealing time, while the out-of-plane direction is a hard magnetic axis (Figure 5b). Furthermore, when

$\varphi -\Delta \varphi =0{}^{\circ},\;\mathrm{the}\;A+M_{0}=M_{R}^{∥}/M_{S}^{∥}$ (the in-plane squareness value from Table 1);$\varphi -\Delta \varphi =90{}^{\circ},\;\mathrm{the}\;M_{0}=M_{R}^{⊥}/M_{S}^{⊥}$ (the value of squareness for out-of-plane direction from Table 1).

The values of Δφ angle are relatively small and close to the value of 0 degrees, confirming that the easy and hard magnetization axis are in-plane and normal to the plane directions, respectively. The calculated values of $A+M_{0}=M_{R}^{∥}/M_{S}^{∥}$ and $M_{0}=M_{R}^{⊥}/M_{S}^{⊥}$ agree very well with the values presented in Table 1.

Table 2 also presents the results of the orientation ratio (OR) parameter defined as 1−M0A+M0 (corresponding to 1−MR⊥/MS⊥MR∥/MS∥). Typically the OR parameter is defined as a ratio of the remanence for the easy magnetization axis to the remanence measured for the hard magnetization direction [65], but here the parameter is restricted to the range between 0 (for isotropic distribution) and 1 (for perfect in-plane magnetic anisotropy) due to the renormalization used. The values show a decrease with annealing time, confirming the more isotropic magnetic behavior for samples annealed for the longest time. In the beginning, the OR value strongly decreases then stabilizes for the annealing time between 4 and 30 min, and further decreases for the longer annealing time. Similar behavior was found for $K_{eff}$ values (see Table 1). The loss of film continuity occurring between 30 and 60 min of annealing reduces the shape anisotropy and, in consequence, leads to the reduction of in-plane anisotropy, resulting in the isotropic distribution of magnetic moments.

To check if the presence of two crystallographic phases of FePd alloy, A1 and L1_0_, can be identified in the magnetic measurement, and determine how they influence the magnetization switching behavior, the switching field ($H_{SF}$) as the first derivative of an hysteresis loop’s branch was calculated. The maximum of dM/dH corresponds to the $M\left(H\right)$ inflexion point, which is the point where changes in magnetization are the fastest and the largest part of the sample is switched. Hence, the $H_{SF}$ is defined as a field when an inflexion of the magnetization loop happens. The width of the dM/dH maxima determines the switching field distribution (SFD). Figure 6 shows an exemplary first derivative for the sample annealed for 60 min with the magnetic field applied at an angle of 40 degrees. The first derivative was fitted with the Lorentz function having an asymmetric distribution with different values of half width at half maxima for the left ($\omega_{L}$) and right ($\omega_{R}$) side (see also Appendix A). A Lorentz-like distribution of switching fields suggests a cooperative behavior of magnetic moments and the presence of magnetic interactions between them. In the case of non-interacting magnetic clusters, Gaussian-like behavior is expected.

The dependence of the switching field as a function of the direction of the applied magnetic field can be used to determine the magnetization reversal mechanism. Figure 7 shows the results of $H_{SF}\left(\varphi \right)$ for different times of annealing, where solid lines are the fitted functions of the modified Kondorsky model (weighted sum of coherent rotation and domain wall motion mechanism, Figure 7a) and the M-type multidomain ferromagnet with rotating magnetization model (Figure 7b); for details, see Appendix A. The shape of $H_{SF}\left(\varphi \right)$ strongly depends on the annealing time. In samples annealed for short time, a strong increase in switching field values around 70–80 degrees can be observed, accompanied by a firm reduction in $H_{SF}$ above these angles. This behavior suggests a dominant role of wall nucleation and motion for small angles, while the rotation of magnetic moments becomes predominant for angles close to the hard magnetic axis. On the other hand, for samples annealed for the longest time, a strong decrease in the $H_{SF}\left(\varphi \right)/H_{SF}\left(0\right)$ value below 1 for angles above 60 degrees is visible, preceded by a very weak maximum. Zhao et al. [66] observed similar angular dependences of the coercive field for L1_0_ FePt alloy and explained the almost flat curves as an effect of domain wall pinning.

Figure 7c presents values of critical angles $\varphi_{C}^{modK}$ and $\varphi_{C}^{M-type}$ corresponding to the maximum for modified Kondorsky and M-type models from Figure 7a,b. The remaining parameters describing modified Kondorsky or M-type reversal mechanisms are collected in Appendix A. A large similarity in values $\varphi_{C}^{modK}$ and $\varphi_{C}^{M-type}$ is observed. A reduction in the critical angles $\varphi_{C}$ with annealing time is found and presented in Figure 7c. In the case of the modified Kondorsky model, the reduction is connected with an increase in an inverse domain wall nucleation field ($H_{0}^{DWM}$) in relation to the magnetic field needed for reversal by magnetization rotation ($H_{0}^{CR}$). The relationship between these two fields (Figure 8a) indicates that the nucleation of inversed domain wall becomes less probable for samples annealed for a longer time and the influence of the rotation mechanism increases with annealing time. The turnover point between the two switching mechanisms happens at angle $\varphi_{C}^{modK}$ when the $H_{SF}\left(\varphi \right)/H_{SF}\left(0\right)$ curve shows a maximum.

On the other hand, in the case of the M-type model, the reduction of the $\varphi_{C}^{M-type}$ angle is connected with a change in the ratio of demagnetization factors $N=\frac{N_{easy-axis}}{N_{hard-axis}+N_{A}}$, i.e., the demagnetization factors for the easy and hard magnetic axis ($N_{easy/hard-axis}$) and the mean magnetocrystalline anisotropy ($N_{A}$) (for details see Appendix A). If N→0 (i.e., when $N_{easy-axis}\to 0$), the reversal model reduces to a simple Kondorsky model (see Appendix A). In the case of the studied samples, the N parameter is different to zero, leading to a mixed case of the Kondorsky mode and reversal by magnetic moment rotation. The observed increase in N parameter with annealing time (see Figure 8a) is a consequence of the reduction in magnetic anisotropy when the $N_{easy-axis}$ increases and $N_{hard-axis}$ decreases. Therefore, the N parameter becomes an important factor that demonstrates which reversal mechanism is dominant, nucleation and motion of domain walls or magnetic moment rotation.

Many aspects could influence the magnetic anisotropy, such as the variations in shape anisotropy caused by changes in film morphology, or alterations in magnetic properties with modification of phase composition affecting the magnetocrystalline anisotropy. SEM analysis clearly showed the loss of film continuity in samples annealed longer than 30 min, where an observed development of islands leads to the reduction of shape anisotropy. On the other hand, an increase in magnetocrystalline anisotropy (and a change of the $N_{A}$ factor) can be expected since an increased contribution of the L1_0_ phase was found in samples subjected to long annealing. In the case of a (111) crystallographic texture, which is dominant in our samples, the magnetocrystalline anisotropy promotes an easy axis at the azimuthal angle of $\pm 54.7^{{}^{\circ}}$ from the film normal.

The linear correlation of $H_{0}^{DWM}/H_{0}^{CR}$ vs. $\varphi_{C}^{modK}$ and N vs. $\varphi_{C}^{M-type}$ is presented in Figure 8b. Such a good agreement between both models demonstrates their complementarity, i.e., both describe a case of a mixed magnetization reversal process between domain wall motion and rotational behavior. Furthermore, both models allowed, with similar accuracy, the determination of the critical angle for which the change of dominant magnetization reversal mechanism happened. The onset of the switching is shifted to lower angles proportionally to an increase in $H_{0}^{DWM}/H_{0}^{CR}$, which describes the energy cost for the nucleation of inversed domain wall and to the N factor related to a reduction in magnetic anisotropy.

The switching field distribution was analyzed to fully understand the switching behavior. The previously mentioned asymmetry of dM/dH curves (Figure 6) suggests the presence of magnetic interparticle interactions, i.e., the strong influence of the L1_0_ phase of the magnetically hard part of the sample onto the magnetically softer parts with a chemically disordered A1 structure.

Figure 9a shows the angular dependence of the switching field distribution width (ω=HWHM) for the left and right sides of the dM/dH peaks. Large differences between the SFD for an easy and a hard magnetization axis are shown. For small angles, ω has a value of around 1 kOe, while for angles close to 90 degrees, an increase above 10 kOe is observed. The increase is very steep and takes place around 60 deg for samples annealed for a short time, while for longer times of annealing, it is more blurred and shifts to lower angles. Additionally, the starting value of ω (around $\varphi \approx 0^{0}$) is slightly higher and reaches 2 kOe for samples annealed for a long time. Therefore, the longer the annealing time, the wider the switching field distribution, meaning a larger magnetic disorder. The main reason is the change in film morphology and progress in the dewetting process, causing the creation of separated aggregates of FePd alloy with a large distribution of sizes. At the same time, this mechanism is responsible for a decrease in magnetic shape anisotropy and a more isotropic behavior of magnetization.

Figure 9b shows the difference Δω between the right $\omega_{R}$ and the left $\omega_{L}$ width of SFD. For smaller values of Δω, the SFD is more symmetric. The smallest values occur for angles φ close to the easy magnetization axis and for samples annealed for a short time. This result suggests that the magnetic shape anisotropy, dominating in those samples, is the main factor responsible for the observed reversal mechanism and distribution of switching fields. For all samples, Δω shows a maximum at angle $\varphi_{SFD}^{\max\Delta \omega }$. The angle shifts from values close to 90 deg for samples annealed for a short time to a value of 30 deg for the sample annealed for 300 min. Figure 9c presents a map of Δω, with additionaly marked angles for which the maximum appears. The values were normalized to the mean value of the sum of the widths from the right and left sides: $\left(\omega_{R}-\omega_{L})/\frac{1}{2}(\omega_{R}+\omega_{L}\right)$. The obtained values of $\varphi_{SFD}^{\max\Delta \omega }$ resemble very well the critical angles found from the switching mechanisms of $\varphi_{C}^{M-type}$ and $\varphi_{C}^{modK}$. Therefore, $\varphi_{SFD}^{\max\Delta \omega }$ is closely related to the change of the magnetization reversal mechanism between domain wall motion and magnetization rotation. A linear correlation dependence of the critical angle and the maximum in SFD asymmetry is presented in Figure 9d.

The SFD is small and symmetric as long as the film is continuous and the shape anisotropy dominates. This suggests a cooperative behavior of magnetic moments within the sample. If most of the material forms a continuous flat film, domain wall propagation in the sample plane direction is easily realized. On the other hand, the formation and propagation of the domain walls normal to the film plane direction, being a hard magnetic axis, is no longer favored, since the typical domain wall width in the FePd alloy has a value of ~7.5 nm (ref. [67]), while the film thickness is 12.5 nm. The progress of dewetting reduces the shape anisotropy and induces the granular structure which shifts the critical angle of the magnetization reversal to lower values. Those changes are the main reason for the increase in SFD, as well as its asymmetry. One of the most important reasons for SFD asymmetry is the interaction between the L1_0_ magnetic hard phase and magnetically softer phases (disordered A1, A_6,_ or other partially ordered phases) randomly distributed and mixed. The hard phase pins the magnetic soft part of the material and leads to behavior similar to one observed in exchange spring systems or exchange-coupled materials. The cooperative behavior between hard and soft magnetic phases is dominated by magnetic shape anisotropy for the samples with continuous morphology, and by the grain size distribution for dewetted samples. The effect of exchange hardening [68] and the enhancement of coercive field resulting from the magnetic exchange interactions between A1 and L1_0_ phases was previously observed in the FePd patterned system of core/shell nanoparticles [69] and dot arrays [70].

## 4. Conclusions

We combined detailed magnetic studies with structural and morphology analysis to inspect the process of phase transition and chemical ordering of FePd alloy accompanied by solid-state dewetting. Thermal treatment was used to fabricate a hard magnetic L1_0_ phase and induce changes in the film morphology from a continuous film to randomly distributed particles of the FePd alloy. Structural studies showed that the FePd is a mixture of chemically disordered A1 and ordered L1_0_ phases. The transformation to the L1_0_ phase was not completed even after 5 h of annealing, and the presence of both phases was observed in all samples. The analysis of the grain growth process of XRD crystallites and SEM particles revealed the importance of solid-state dewetting as a crucial boosting parameter of the transformation from the A1 to the L1_0_ phase. We concluded that the continuous film is composed mostly of a disordered A1 phase, whereas the island-patterned structures consist mostly of the L1_0_ phase. The crucial time is found to be approximately 30 min of annealing, which is when the film loses its continuity and the solid-state dewetting strengthens the diffusion and stimulates the transformation and growth of L1_0_ grains.

We performed an $M\left(H,\varphi \right)$ study and analyzed the changes in magnetic anisotropy, magnetization reversal mechanism and switching field distribution. They showed that, again, the critical time point occurs at about 30 min of annealing when a loss of film continuity and the formation of randomly distributed aggregates of different sizes occur. The change in film morphology reduces the magnetic shape anisotropy, observed as the decrease in the orientation ratio. The effect leads to the almost isotropic orientation of magnetization in samples annealed for 60 min or longer.

The angular-dependent switching behavior of magnetization was found to be a mixed process of reversal by domain wall motion and magnetization rotation, where the reversal dominated by domain wall motion was observed for angles close to the film plane, while reversal dominated by rotation was found for angles close to the perpendicular direction. The critical angle for the change between magnetization reversal mechanisms shifts to lower values with the annealing time. Two models of modified Kondorsky and M-type ferromagnet with a domain structure were applied, both giving similar results. The strongest change was observed for the film with lost continuity and an ensemble of FePd isolated islands. Solid-state dewetting and the formation of island-like FePd structures promote the switching mechanism by rotation of magnetic moments and hinder the domain wall motion. This effect is also connected with the pinning of the domain walls, seen as a flattening of the switching field angular dependence and an increase in magnetic field coercivity. The coexistence of different phases and grains with large size distribution is visible in SFD, which reaches 10 kOe for directions close to the hard magnetic axis. The appearance of large asymmetry between the right and left side of the switching field distribution, i.e., for magnetic fields larger and lower then $H_{sf}$, revealed a presence of strong exchange coupling and spring magnet-like behavior with pinning between soft and hard magnetic phases. The behavior of the SFD asymmetry, and, hence, magnetic coupling, correlate with a change in magnetic reversal behavior and sample morphology, showing that the magnetic pinning is governed by the shape anisotropy and distribution of grain sizes. For a continuous film with strong magnetic shape anisotropy, the material behaves collectively with a small asymmetry of SFD for angles close to the easy axis. For a direction close to the hard magnetic axis, the situation changes and magnetic interactions are ineffective because of the external magnetic field, leading to large randomness in reversal behavior. After annealing, the switching behavior and SFD in all directions are more alike, which results from a large grain distribution rather than magnetic anisotropy.

## Figures and Tables

**Figure 1 materials-16-00092-f001:**
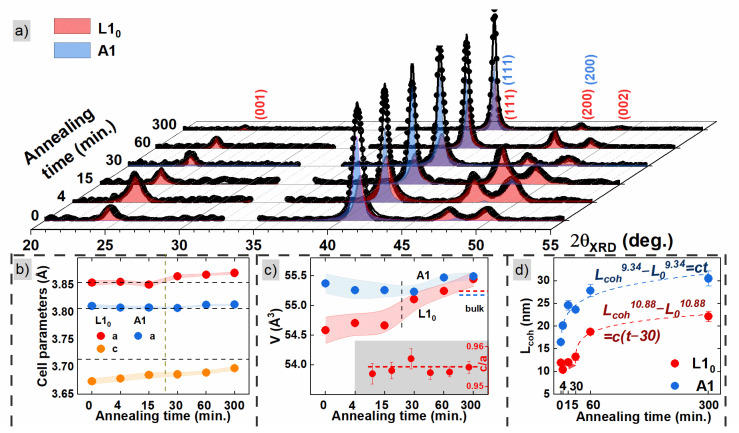
(**a**) Diffractograms of FePd thin alloy films after different times of annealing. The blue and red colors correspond to the A1 phase and the L1_0_ phase, respectively. The parameters extracted from the XRD patterns are presented below: (**b**) cell parameters, (**c**) cell volume, and (**d**) coherence length. The inset of figure (**c**) shows the tetragonal distortion ratio c/a.

**Figure 2 materials-16-00092-f002:**
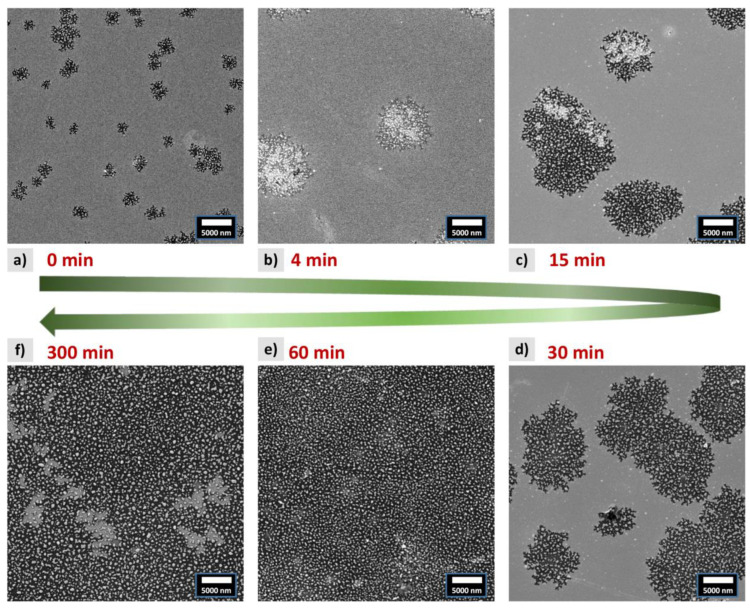
SEM studies of morphology evolution as a function of annealing time for the FePd alloy thin film.

**Figure 3 materials-16-00092-f003:**
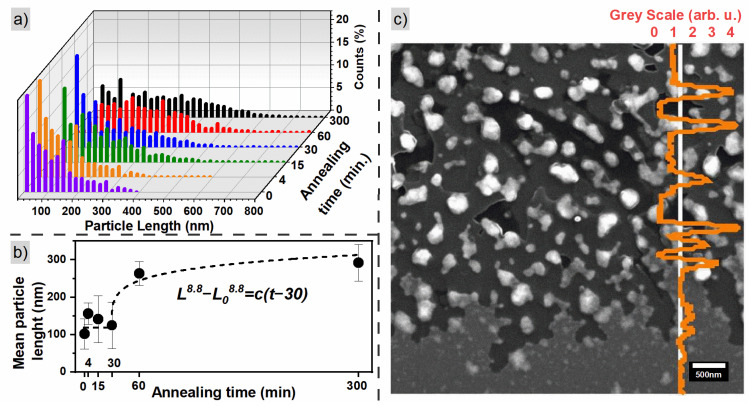
(**a**) Histogram of particle sizes from SEM microscopy fitted with gamma distribution, different colors correspond to different times of annealing; (**b**) particle size mean values; (**c**) SEM image at the void edge of the sample annealed for 4 min showing successive changes in film morphology. The orange line on the right side of the image shows the variation in the grayscale of the SEM signal, where 0 corresponds to the darkest regions, i.e., uncovered silica substrate, and 1 corresponds to the regions of unchanged FePd thin film.

**Figure 4 materials-16-00092-f004:**
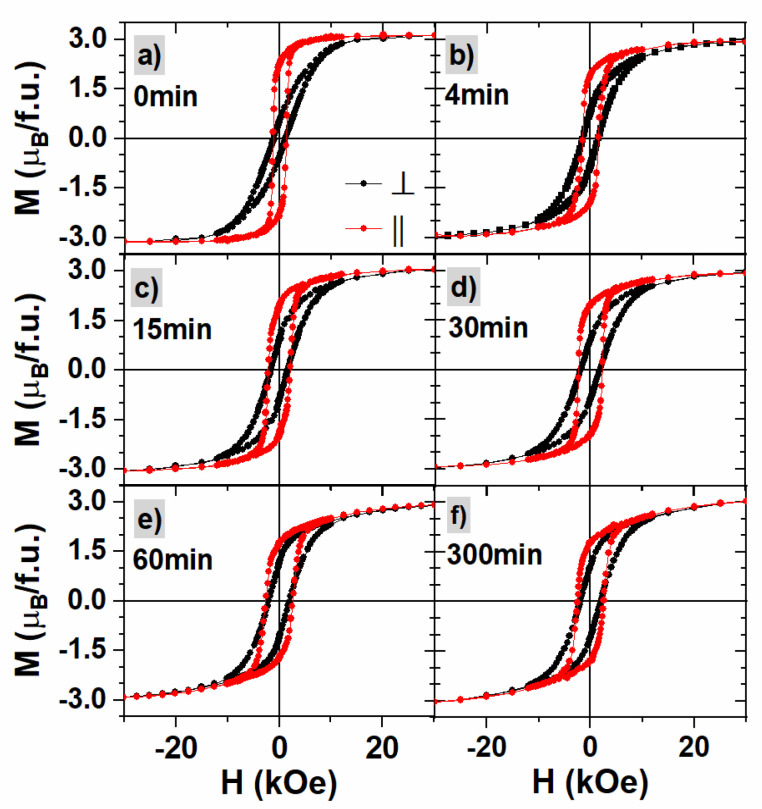
Magnetic hysteresis loops for annealed samples measured with the magnetic field perpendicular (black curves) and parallel (red curves) to the sample surface. Graphs (**a**–**f**) refer to different times of annealing indicated in the left top corner.

**Figure 5 materials-16-00092-f005:**
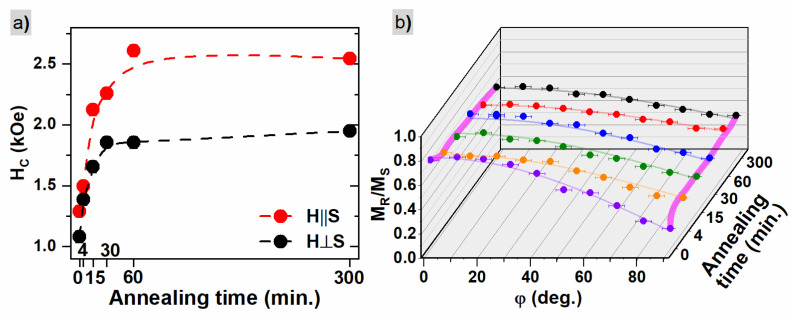
(**a**) Coercive field $H_{C}$ vs. annealing time for in-plane and out-of-plane geometries and (**b**) angular dependence of the $M_{R}/M_{S}$ ratio where different colors are for samples annealed with different time.

**Figure 6 materials-16-00092-f006:**
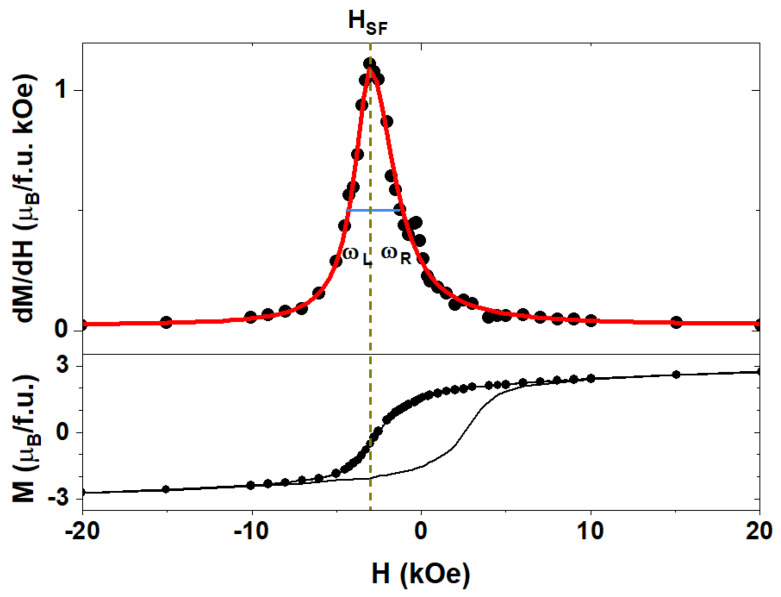
Switching field (top) and $M\left(H\right)$ curves (bottom) for the sample annealed for 60 min with the magnetic field applied at an angle of 40 degrees. The red line is a fit with asymmetric Lorentz function.

**Figure 7 materials-16-00092-f007:**
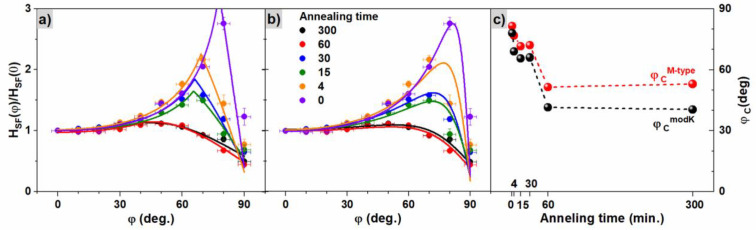
Angular dependence of switching field for samples annealed for different times with solid lines corresponding to fitted functions of (**a**) the modified Kondorsky model and (**b**) the M-type model of the multidomain ferromagnet. (**c**) Critical angle of reversal magnetization mechanism $\varphi_{C}^{M-type}$ and $\varphi_{C}^{modK}$.

**Figure 8 materials-16-00092-f008:**
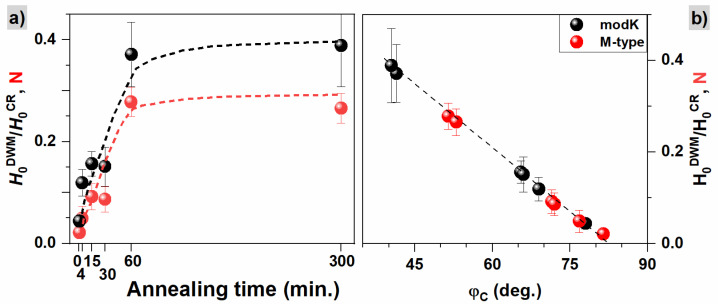
(**a**) Ratio between the nucleation field of the inverse domain wall and the rotation field of the magnetization vector, $H_{0}^{DWM}/H_{0}^{CR}$, in the modified Kondorsky model and ratio of demagnetization factors N in the M-type model as a function of annealing time. (**b**) Correlation of the $H_{0}^{DWM}/H_{0}^{CR}$ parameter with the critical angle $\varphi_{C}^{modK}$ in the modified Kondorsky model and the ratio of demagnetization factors N with $\varphi_{C}^{M-type}$ in the M-type model.

**Figure 9 materials-16-00092-f009:**
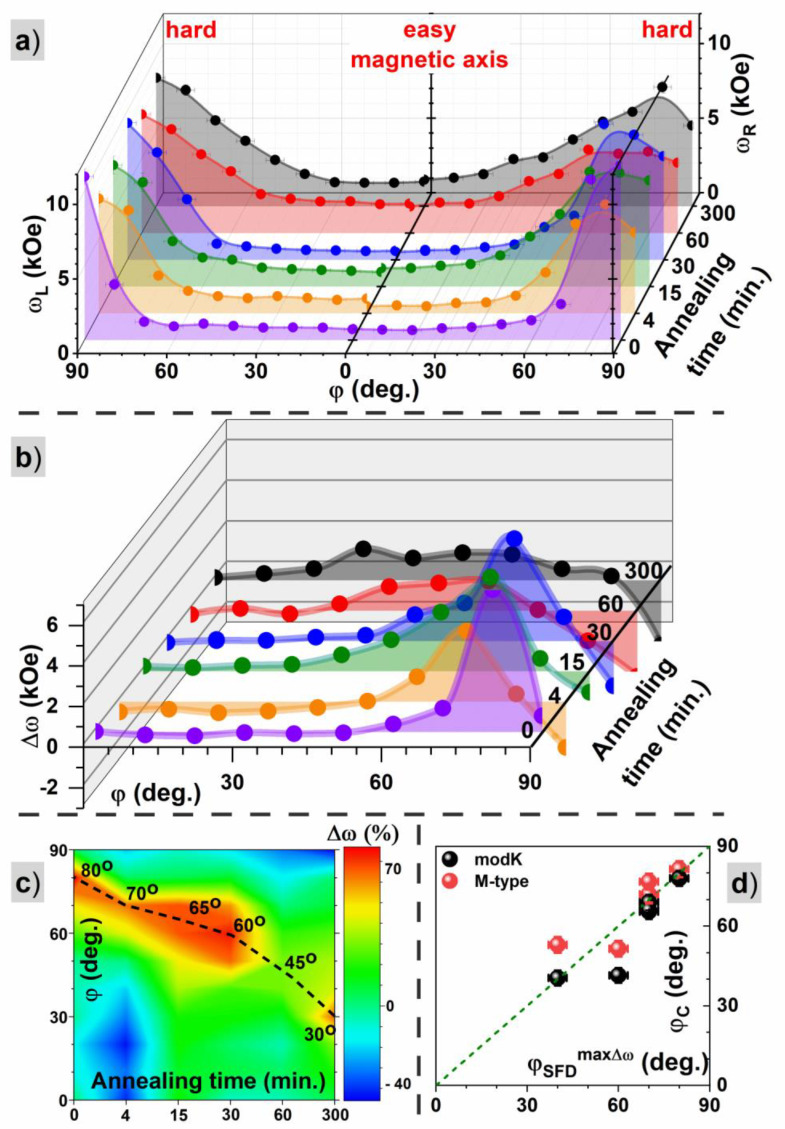
(**a**) Switching field distribution for samples treated with different times of annealing. The $\omega_{L}$ and $\omega_{R}$ are the widths of the left and right sides of the switching field distribution. (**b**) Difference Δω between the right $\omega_{R}$ and left $\omega_{L}$ width of the switching field distribution at different stages of annealing. (**c**) Map of Δω data with colors corresponding to the renormalized Δω value. The dashed line marks angles where the maximum of the Δω is observed. (**d**) Correlation of the critical angle of the reversal mechanism and the angle of maximal asymmetry in the SFD. The colors on (**a**,**b**) refer to samples annealed for different times.

**Table 1 materials-16-00092-t001:** Coercive field $H_{C}$, remanence $M_{R}$ and saturation magnetization $M_{S}$, loop squareness $M_{R}/M_{S}$ for in-plane and out-of-plane measurements, and effective energy density anisotropy constant $K_{eff}$.

Time of Annealing (min)	$H_{C}^{\left|\right|}$ (kOe) ±0.05	$M_{R}^{\left|\right|}$ ($\mu_{B}/$f.u.) ±0.03	$M_{S}^{\left|\right|}$ ($\mu_{B}/$f.u.) ±0.03	$M_{R}^{\left|\right|}/M_{S}^{\left|\right|}$ ±0.02	$H_{C}^{⊥}$(kOe) ±0.05	$M_{R}^{⊥}$ ($\mu_{B}/$f.u.) ±0.03	$M_{S}^{⊥}$ ($\mu_{B}/$f.u.) ±0.03	$M_{R}^{⊥}/M_{S}^{⊥}$ ±0.01	$K_{eff}$ (MJ/m^3^) ±0.02
0	1.29	2.28	3.11	0.73	1.08	0.53	3.12	0.17	0.41
4	1.49	1.91	2.97	0.64	1.38	0.81	2.96	0.27	0.26
15	2.12	1.92	3.06	0.63	1.66	0.91	3.04	0.30	0.26
30	2.26	1.94	2.94	0.66	1.86	0.89	2.96	0.30	0.26
60	2.61	1.69	2.88	0.60	1.85	1.15	2.94	0.39	0.16
300	2.55	1.79	3.08	0.61	1.95	1.08	3.09	0.35	0.16

**Table 2 materials-16-00092-t002:** Parameters of the $M_{R}/M_{S}\left(\varphi \right)$ dependence fitted with Equation (2), together with the calculated orientation ratio OR.

Time of Annealing (min)	A	$M_{0}$ $\left(=M_{R}^{⊥}/M_{S}^{⊥}\right)$	$A+M_{0}$ $\left(=M_{R}^{∥}/M_{S}^{∥}\right)$	Δφ (deg.)	OR $\left(1-\frac{M_{0}}{A+M_{0}}\right)$
0	0.613(22)	0.151(14)	0.764(36)	1.7	0.80(4)
4	0.399(14)	0.249(9)	0.648(24)	3.0	0.62(4)
15	0.361(14)	0.279(9)	0.641(23)	3.2	0.56(5)
30	0.379(12)	0.283(8)	0.662(20)	2.8	0.57(4)
60	0.216(5)	0.377(4)	0.593(9)	3.5	0.36(3)
300	0.256(9)	0.331(6)	0.588(14)	4.4	0.44(4)

## Data Availability

The datasets generated and analyzed during the current study are available from the corresponding author upon reasonable request.

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
