# Peer review of "Solid-State Dewetting as a Driving Force for Structural Transformation and Magnetization Reversal Mechanism in FePd Thin Films"

_materials, 2022, doi:10.3390/ma16010092_

Round 1

Reviewer 1 Report

This manuscript investigates the influence of solid state dewetting process on structure transition and magnetization reversal mechanism in FePt system. This informative research is importance of the controlled preparation technique of FePt with the just-right morphology, structure and magnetic properties by utilizing thermal evaporation method. However, some minor issues should be addressed properly before this manuscript is accepted.

(1)  How to determine the composition of FePd film with Fe45Pd55? Is energy dispersive spectroscopy? What is the role of solid state dewetting in film composition and thickness? In addition, it’s better to introduce the substrate in Part 2 (Materials and Methods).

(2)  The amount of the L10 phase is non-monotonous with increasing annealing time, and the tetragonal distortion ratio distinctly changes, especially for T=15 minutes. These phenomena seem not to be originated from fitting error. Please provide the explanations.

(3)  What about the roughness of FePd films with different annealing temperatures? If the film loses its continuity, its application would be limited, especially for spintronics and electrical transport.

(4)  Page8 Line268: In general, the positive value of anisotropy energy density (Keff) represents that the easy axis of film is perpendicular to its surface, whereas the easy axis in this manuscript is in-plane. Please check the sign of Keff.

(5)  There is lack of some discussions on the magnitude of magnetocrystalline anisotropy-induced demagnetizing factor and a quantitative comparison with shape anisotropy.

(6)  The labels of crystallographic planes and switching field distribution width should be added into Figure 1(a) and Figure 6, respectively.

(7)  Page11 Line363: The “Neasy-ax” should be “Neasy-axis”.

(8)  Please note the unit/number format and check the grammatical, spelling and punctuation mistakes. For example, “℃/min” could be “℃/min”; “℃” could be “℃”; “L10” could be “L10” (Page3 Line127); there is lack/excess of a comma (Page8 Line276; Page8 Line288; Page9 Line315; Page11 Line353); please check the label of vertical coordinates in Figure 3(b); “~7,5 nm” could be “~7.5 nm” (Page14 Line443).

Reviewer 2 Report

accepted in present form

Author Response

We thank the reviewer for his/her work and positive opinion about the manuscript

Reviewer 3 Report

Comments

The manuscript “Solid state dewetting as a driving force for structural transformation and magnetization reversal mechanism in FePd thin film” is of interest for the MDPI journal “Materials”. The paper is studying microstructure and properties of FePd thin film. Nevertheless, in order to be considered for publication, some improvements need to be performed:

1. The English writing level is lower and needs to be further modified sentence by sentence;

2. The voice in the abstract is suggested to be modified, e.g: We studied the morphology, structure and magnetic properties... . It should be modified as morphology, structure and magnetic properties were studied.

3. Annealing process set in the experiment: 4h,15 h,30 h,60 h,300 h, why set such a time interval?

4. As shown in Fig. 1, the scale spacing of the XRD pattern is too long, it is suggested to be lower. All “Figure.X” in the text should be abbreviated as “Fig.X”

5. The conclusions are too long and not concise enough. Please modify them.

6.The analysis of magnetization reversal mechanism  in FePd thin film wasn’t clear according to different annealing process.

Reviewer 4 Report

In this manuscript, the authors study the process of solid state dewetting in FePd thin films and how it affects the film's structural and magnetic properties. For structural properties, authors present cell parameters, morphology evolution and particle size as a function of annealing time. For magnetic properties, authors show how magnetic hysteresis loops, coercive field and other parameters will change as a function of annealing time and angular dependence. It's a complete work given the adequate experimental results, and I am favorable to the publication of this manuscript.

Author Response

We thank the reviewer for his/her work and positive opinion about the manuscript.